# Neuroprotective Effect of Membrane-Free Stem Cell Extract against Amyloid Beta $_{25-35}$-Induced Neurotoxicity in SH-SY5Y Cells

**Hye Sook Park [1,2,†], Qi Qi Pang [1,†], Young Sil Kim [2], Ji Hyun Kim [3,\*] and Eun Ju Cho [1,\*]**

1. Department of Food Science and Nutrition, Pusan National University, Busan 46241, Korea; cando1226@naver.com (H.S.P.); pangqq@pusan.ac.kr (Q.Q.P.)
2. T-Stem Co., Ltd., Changwon 51573, Korea; tstem@t-stem.com
3. Department of Food Science, Gyeongsang National University, Jinju 52725, Korea
* Correspondence: jihyunkim@gnu.ac.kr (J.H.K.); ejcho@pusan.ac.kr (E.J.C.)
† Authors with equal contributions.

**Abstract:** Amyloid beta (Aβ) produced by the amyloidogenic pathway induces neurotoxicity, and its accumulation is a well-known cause of Alzheimer's disease (AD). In this study, the protective effect of membrane-free stem cell extract (MFSCE) derived from adipose tissue against Aβ$_{25-35}$-induced neurotoxicity in the neuronal cells was investigated. Treatment with MFSCE increased cell viability and decreased lactate dehydrogenase (LDH) release in a dose-dependent manner, compared with the Aβ$_{25-35}$-induced group. The level of reactive oxygen species (ROS) was significantly increased in neuronal cells induced by Aβ$_{25-35}$, whereas MFSCE treatment dose-dependently reduced ROS production. Treatment with MFSCE attenuated neuroinflammation and neuronal apoptosis by down-regulating inducible nitric oxide synthase, cyclooxygenase-2, and B-cell lymphoma 2-associated X protein in treated SH-SY5Y cells induced by Aβ$_{25-35}$. Furthermore, MFSCE significantly downregulated the expression of the amyloidogenic pathway-related proteins, such as amyloid precursor protein, β-secretase, preselin-1, and preselin-2. Therefore, this study indicated a neuroprotective effect of MFSCE against neurotoxicity induced by Aβ$_{25-35}$, suggesting that it is a useful strategy for the treatment of AD.

**Keywords:** membrane-free stem cell extract; neuroinflammation; Alzheimer's disease; reactive oxygen species; amyloid beta





## 1. Introduction

Alzheimer's disease (AD) is characterized by progressive cognitive and memory deficits, and is among the most common public health problems worldwide [1]. Abnormal accumulation of amyloid beta (Aβ) is widely accepted as a key driving factor for the pathogenesis of AD in the brain [2]. In the amyloidogenic pathway, Aβ is produced from the cleavage of the amyloid precursor protein (APP) via the activation of β- and γ-secretase in the brain [3]. Aβ accumulates in the extracellular region and forms amyloid plaque, which leads to neurotoxicity in patients with AD [4]. In addition, Aβ induces several negative effects, such as neuroinflammation, oxidative stress, and neuronal apoptosis [5]. Oxidative stress induced by the over-production of reactive oxygen species (ROS) stimulates neuroinflammation via releasing inflammatory mediators and cytokines [6]. Furthermore, oxidative stress and neuroinflammation lead to neuronal apoptosis via the upregulation of pro-apoptotic factors [5,7]. Therefore, the regulation of oxidative stress and neuroinflammation is crucial in the inhibition of Aβ progression.

Stem cells are undifferentiated cells that maintain homeostasis through differentiation and regeneration of cells in the body [8]. Adult stem cells have the ability to differentiate into multiple cell types and possess anti-inflammatory activity; therefore, they are considered for the treatment of heart disease, diabetes mellitus, and others [9]. Recently,

many studies have focused on the effects of stem cells for the treatment of AD. Adult stem cells improve cognitive impairment and memory deficits by reducing Aβ deposition and enhancing the rate of Aβ clearance in an AD mouse model [10,11]. However, adult stem cells used for the treatment for AD encounter high cellular immune rejection by the human immune system due to the presence of antigens on their membranes, resulting in an extremely high economic burden due to their usage [12,13].

Membrane-free stem cell extract (MFSCE) is produced by the removal of membranes from adult stem cells using patented technology to overcome the disadvantage of cellular immune rejection [14]. MFSCE is composed of components of adult stem cells from human adipose tissue, so it is associated with several biological activities, such as metabolic processes and biological regulation [14]. In previous studies, we demonstrated several beneficial effects of MFSCE [14,15]. The MFSCE attenuated inflammatory reaction by decreasing nitric oxide (NO) production in lipopolysaccharide (LPS)-induced RAW264.7 cells [14]. The MFSCE indicated antioxidant activity via the elevation of antioxidant proteins in LLC-PK$_1$ cells [15]. However, the neuroprotective effect of MFSCE against neurotoxicity has not yet been investigated. In the present study, the protective effect of MFSCE against neurotoxicity in neuronal cells induced by Aβ was investigated. To understand the mechanisms underlying the neuroprotective effect of MFSCE, factors related to oxidative stress, neuroinflammation, neuronal apoptosis, and amyloidogenic pathway were evaluated in neuronal cells.

## 2. Materials and Methods

### 2.1. Preparation of MFSCE

The MFSCE was obtained from T-Stem Co. (Changwon, Korea). Briefly, human adipose tissues were separated and purified, and the extracted cells were cultured in a serum-free cell culture medium in an incubator with 5% $CO_2$ at 37 °C. The donor of human adipose tissue was a healthy female in her twenties with 2-degree obesity. The adipose tissue was obtained after the blood tests, which were proven to be appropriate. When cells reached confluence, they were subcultured. The cells were collected and ultrasonicated to remove their membranes. Next, the debris from the cells was removed by centrifugation at 800–1500 g, following successive filtration. MFSCE, the final product, was proven to be a non-toxic substance through 9 safety tests performed by the Good Laboratory Practice accreditation authority [14].

### 2.2. Reagents

Aβ$_{25–35}$ was purchased from Sigma-Aldrich (St. Louis, MO, USA), and it was incubated at 37 °C for 72 h in saline solution. After aggregation of Aβ$_{25–35}$, we used it in experiments. 3-(4,5-Dimethyl-2-thiazolyl)-2,5-diphenyl-2H-tetrazolium bromide (MTT) and dichlorofluorescein diacetate (DCF-DA) were obtained from Bio Pure (Ontario, Canada) and Sigma-Aldrich (St. Louis, MO, USA), respectively. Dulbecco's modified Eagle's medium (DMEM), penicillin-streptomycin, fetal bovine serum (FBS), and trypsin-EDTA were purchased from Welgene (Daegu, Korea). A lactate dehydrogenase (LDH) cytotoxicity detection kit was purchased from Takara Bio (Shiga, Japan). For Western blotting, radioimmunoprecipitation assay (RIPA) solution was obtained from Elpics Biotech (Daejeon, Korea), and enhanced chemiluminescence (ECL) substrate solution from Bio-Rad Laboratories (Clarity Western ECL Substrate kit, Bio-Rad Laboratories, Inc Hercules, CA, USA). Polyvinylidene fluoride membrane was obtained from Millipore Co. (Billerica, MA, USA). Primary and secondary antibodies are as follows: β-actin (1:1000), presenilin 1 (PS1, 1:1000), presenilin 2 (PS2, 1:1000), β-site APP-cleaving enzymes (BACE, 1:100), Bcl-2 associated X (Bax, 1:1000), and anti-rabbit IgG, HRP-linked antibody (1:500) were purchased from Cell signaling Tech., (Beverly, USA). Cyclooxygenase (COX-2, 1:500) and inducible nitric oxide synthase (iNOS, 1:1000) were purchased from Calbiochem Co., (San Diego, CA, USA). APP (1:1000) and B-cell lymphoma 2 (Bcl-2, 1:1000) were purchased from Sigma Aldrich, (St. Louis, MO, USA) and Abcam (Cambridge, MA, USA), respectively.

### 2.3. Cell Culture

The SH-SY5Y neuronal cell was obtained from the Korean Cell Line Bank (Seoul, Korea). Cells were cultured in DMEM containing 10% FBS and 1% penicillin-streptomycin and were maintained in an incubator with 5% $CO_2$ at 37 °C. The cells were subcultured with 0.05% trypsin-EDTA at confluence.

### 2.4. Cell Treatment

The SH-SY5Y cells were seeded at a cell density of $2.5 \times 10^5$ cells/well in a 96 well microplate and incubated for 24 h. The MFSCE was added to the each well at different concentrations (0.5–5 µg/mL) and incubated for 4 h. And then, the cells were treated with 25 µM A$\beta_{25-35}$, followed by incubation for 24 h.

### 2.5. Cell Viability

Cell viability was determined by MTT colorimetric assay [16]. MTT (5 mg/mL) was added to wells. Then, cells were incubated further for 4 h. To solubilize the incorporated formazan crystals, the cell culture medium of each well was replaced with dimethyl sulfoxide. The absorbance was measured at a wavelength of 540 nm using a microplate reader (Thermo Fisher Scientific Inc., Vantaa, Finland).

### 2.6. LDH Release

LDH release was assessed by an LDH cytotoxicity assay using a detection kit [17]. The supernatant of the cells was mixed with an LDH reaction mixture at 1:1 ratio. This mixture was incubated at 25 °C for 30 min, and the absorbance was measured at 490 nm using a microplate reader.

### 2.7. Production of ROS

ROS production was carried out by DCF-DA assay [18]. Cells were incubated with 80 µM DCF-DA solution and incubated for 30 min at 37 °C. Fluorescence was measured at an excitation wavelength of 480 nm and emission wavelength of 535 nm using FLUO star OPTIMA (BMG Labtech, Ortenberg, Germany).

### 2.8. Protein Expression

The SH-SY5Y cells were scraped and lysed with RIPA solution. Proteins were quantitated using a Bio-Rad assay kit [19]. Equal protein samples were separated by 10% or 13% sodium dodecyl sulfate-polyacrylamide gel (SDS-PAGE) and transferred to a membrane by electrophoresis. The membranes were blocked with 5% skim milk at room temperature for 1 h and then incubated with the primary antibody at 4 °C overnight. The membrane was washed three times with PBS-T for 10 min, and then incubated with the secondary antibody at room temperature for 1 h. Again, the membrane was washed three times with PBS-T for 10 min, and then reacted with ECL solution. Protein expression was confirmed using a chemiluminescence imaging system (Davinch-Chemi[TM], Davinch-K, Seoul, Korea). Protein density was analyzed using ImageJ software (National Institutes of Health, Bethesda, MD, USA). The protein level was expressed as a ratio of the band intensity divided by that of the housekeeping protein, β-actin.

### 2.9. Statistical Analysis

All data are presented as mean ± standard deviation (SD). The normal distribution of data was investigated using the Shapiro–Wilk test. When the group means of the data showed normal distribution, the results were analyzed by one-way analysis of variance (ANOVA) followed by Duncan's multiple range test. If the group means of the data showed no normal distribution, the differences between the groups were analyzed using the Mann–Whitney U-test. Statistical significance was set at $P < 0.05$. Each experiment was carried out in triplicate ($n = 3$).

## 3. Results

### 3.1. Effect of MFSCE on Viability of SH-SY5Y Cells Treated with $A\beta_{25-35}$

To evaluate the neuroprotective effect of MFSCE, we measured the cell viability (Figure 1). We observed a significant decrease in cell viability to 63.76% in $A\beta_{25-35}$-treated cells. However, MFSCE-treated cells at concentrations of 0.5, 1, 2.5, and 5 μg/mL significantly increased cell viability to 80.46%, 85.48%, 86.80%, and 87.02%, respectively. This result indicates that MFSCE elevated cell viability of SH-SY5Y cells induced by $A\beta_{25-35}$.

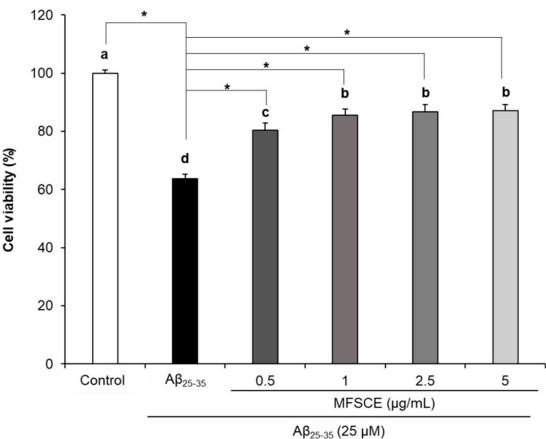

**Figure 1.** Effect of membrane-free stem cell extract (MFSCE) on viability of SH-SY5Y cells treated with $A\beta_{25-35}$. Values are presented as mean $\pm$ SD ($n = 6$). The different letters (a–d) among groups represent significant differences determined by ANOVA and Duncan's multiple range test ($p < 0.05$). * $p < 0.05$ vs. $A\beta_{25-35}$-treated cells.

### 3.2. Effect of MFSCE on Lactate Dehydrogenase (LDH) Activity in SH-SY5Y Cells Treated with $A\beta_{25-35}$

We investigated the effect of MFSCE on LDH activity (Figure 2). LDH activity was 137.30% in $A\beta_{25-35}$-treated cells, compared with 100.00% in the control group. On the other hand, MFSCE-treated cells dose-dependently inhibited LDH activity, compared with the $A\beta_{25-35}$-treated group. LDH activity at MFSCE doses of 0.5, 1, 2.5, and 5 μg/mL was 126.82%, 119.48%, 109.76%, and 100.90%, respectively. This indicates that MFSCE inhibited LDH activity in neuronal cells induced by $A\beta_{25-35}$.

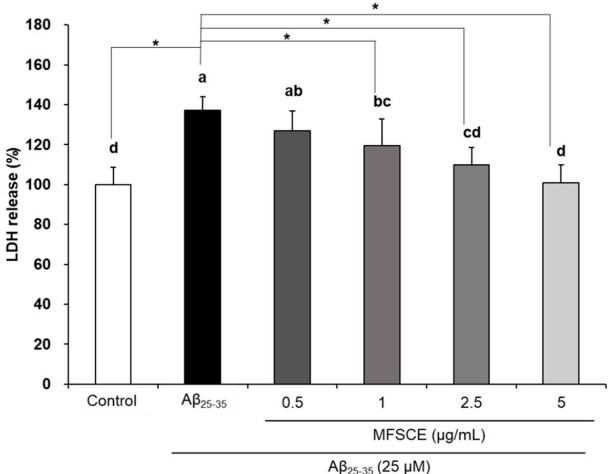

**Figure 2.** Effect of membrane-free stem cell extract (MFSCE) on lactate dehydrogenase release activity in SH-SY5Y cells treated with $A\beta_{25-35}$. Values are presented as mean $\pm$ SD ($n = 6$). The different letters (a–d) among groups represent significant differences determined by ANOVA and Duncan's multiple range test ($p < 0.05$). Asterisk (*) indicates significant differences using Mann–Whitney U-test ($p < 0.05$).

### 3.3. Effect of MFSCE on ROS Production in SH-SY5Y Cells Treated with Aβ25–35

The effect of MFSCE on ROS production is shown in Figure 3. ROS levels were higher in the Aβ25–35-treated group (121.24%) than those of the control group (100.00%). However, when the cells were treated with 0.5, 1, 2.5, and 5 µg/mL of MFSCE, ROS production significantly decreased to 103.89%, 102.29%, 102.99%, and 102.46%, respectively, compared with Aβ25–35-treated cells. Therefore, MFSCE exerted neuroprotective activity against Aβ25–35 by reducing the levels of ROS production.

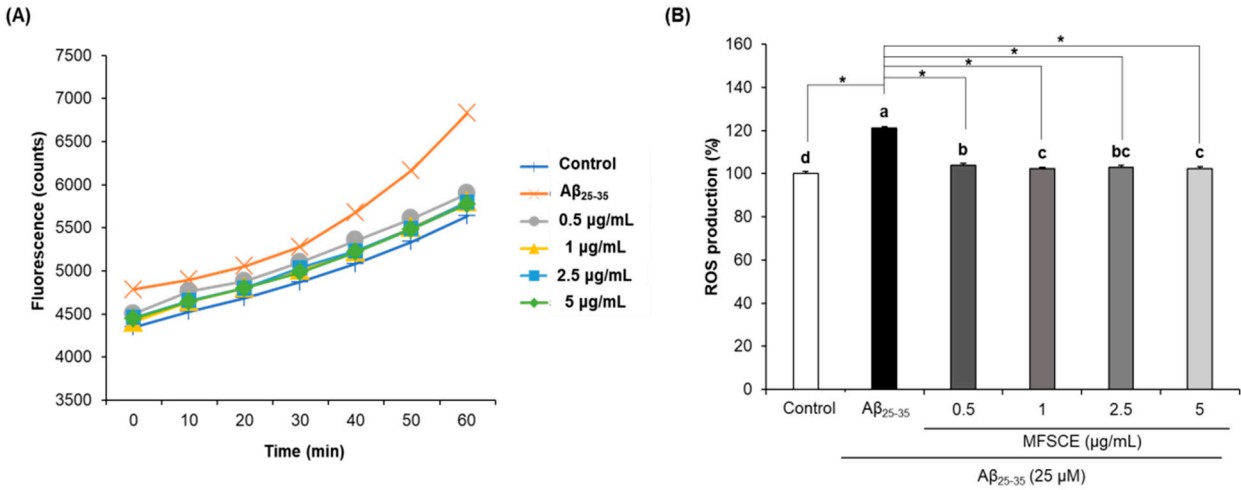

**Figure 3.** Effect of membrane-free stem cell extract (MFSCE) on production of reactive oxygen species in SH-SY5Y cells treated with Aβ25–35. (**A**) Changes in intensity of ROS fluorescence during 60 min. (**B**) The production of ROS in cells treated with different concentrations of MFSCE at 60 min. Values are presented as mean ± SD (*n* = 6). The different letters (a–d) among groups represent significant differences determined by ANOVA and Duncan's multiple range test (*p* < 0.05). * *p* < 0.05 vs. Aβ25–35-treated cells.

### 3.4. Effect of MFSCE on Neuroinflammation in SH-SY5Y Cells Treated with Aβ25–35

To evaluate the protective mechanism of MFSCE against neuroinflammation, we measured the levels of inflammation-related proteins (Figure 4). Treatment with Aβ25–35 upregulated iNOS and COX-2 in SH-SY5Y cells, compared to that in the control cells. On the other hand, cells treated with MFSCE at concentrations of 0.5, 1, 2.5, and 5 µg/mL dose-dependently downregulated iNOS and COX-2. Therefore, this result suggests that the neuroinflammatory effect of MFSCE on neurotoxicity is related to the downregulation of iNOS and COX-2.

### 3.5. Effect of MFSCE on Apoptosis in SH-SY5Y Cells Treated with Aβ25–35

To investigate the effect of MFSCE on neuronal apoptosis, protein levels of Bax and Bcl-2 were measured. The expression of the pro-apoptotic protein, Bax, was upregulated in Aβ25–35-treated cells compared with that in the control cells (Figure 5). However, treatment with MFSCE inhibited Bax protein expression. In addition, the expression of Bcl-2, an anti-apoptotic factor, was significantly inhibited in Aβ25–35-treated cells, while MFSCE-treated cells upregulated the expression of Bcl-2. Furthermore, MFSCE-treated groups showed a dose-dependent decrease in the Bax/Bcl-2 ratio, compared with that of Aβ25–35-treated cells. In particular, the group treated with 5 µg/mL of MFSCE showed a higher extent of decrease in the Bax/Bcl-2 ratio than the groups treated with other concentrations. Therefore, this result demonstrates that MFSCE exerts protective effects against neuronal apoptosis via the downregulation of Bax and upregulation of Bcl-2 protein expressions.

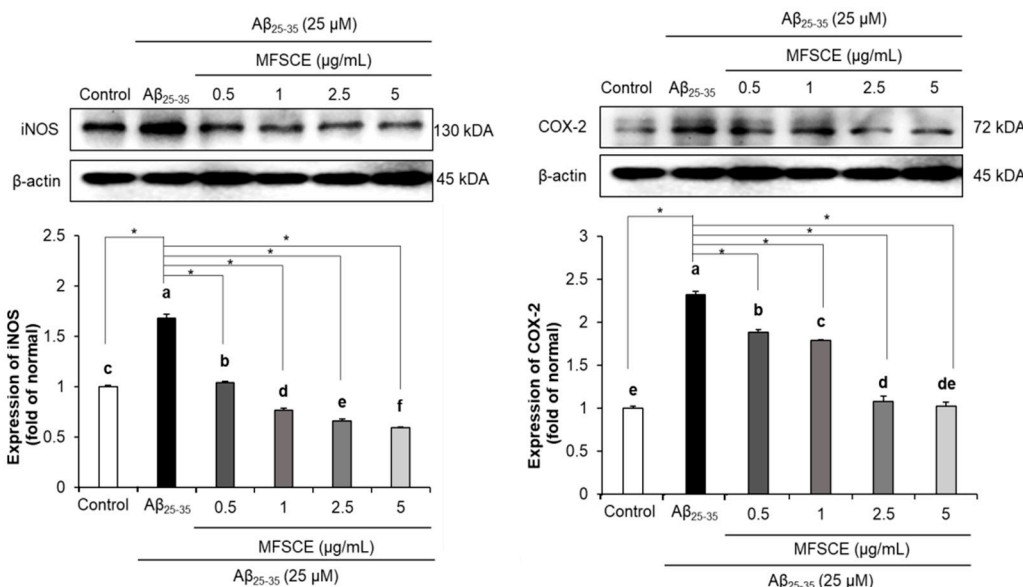

**Figure 4.** Effect of membrane-free stem cell extract (MFSCE) on neuroinflammation-related protein expression in SH-SY5Y cells treated with $A\beta_{25-35}$. Values are presented as mean $\pm$ SD ($n$ = 6). The different letters (a–f) among groups represent significant differences determined by ANOVA and Duncan's multiple range test *(p < 0.05)*. * *p < 0.05* vs. $A\beta_{25-35}$-treated cells. $\beta$-actin was used as a loading control.

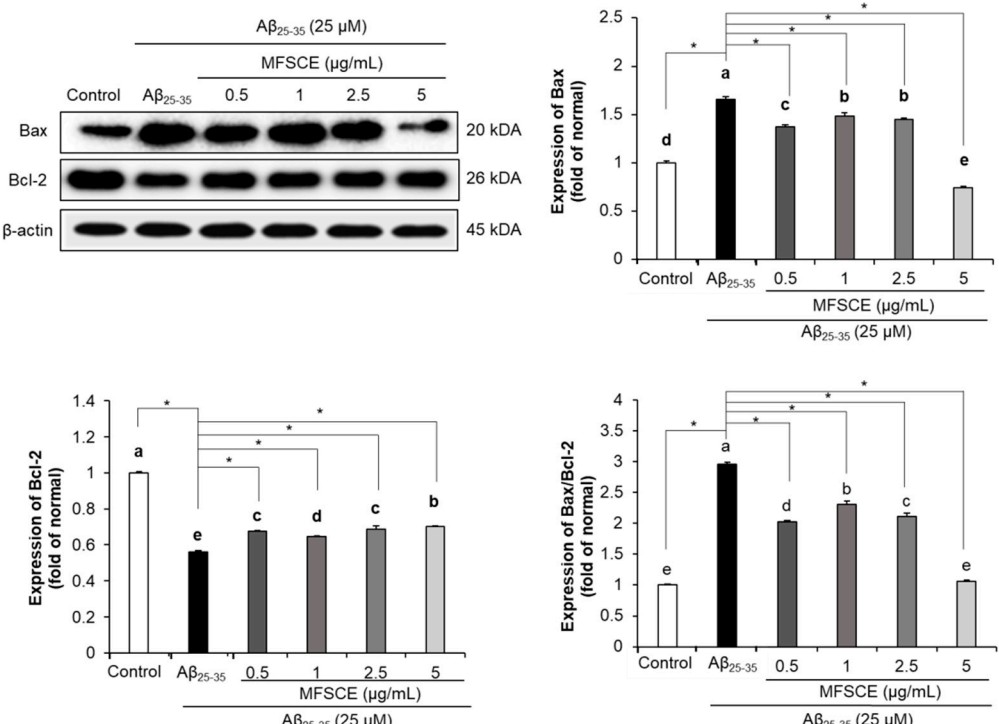

**Figure 5.** Effect of membrane-free stem cell extract (MFSCE) on neuronal apoptosis-related protein expression in SH-SY5Y cells treated with $A\beta_{25-35}$. Values are presented as mean $\pm$ SD ($n$ = 6). The different letters (a–e) among groups represent significant differences determined by ANOVA and Duncan's multiple range test *(p < 0.05)*. * *p < 0.05* vs. $A\beta_{25-35}$-treated cells. $\beta$-actin was used as a loading control.

*3.6. Effect of MFSCE on Amyloidogenic Pathway in SH-SY5Y Cells Treated with Aβ25–35*

To demonstrate the effect of MFSCE on amyloidogenesis, the levels of proteins related to the amyloidogenic pathway were measured. The expression of amyloidogenic proteins, namely, APP, BACE, PS1, and PS2, was significantly upregulated in Aβ25–35-treated cells, compared with that in the control cells (Figure 6). However, treatment with MFSCE inhibited the protein expression of APP, BACE, PS1, and PS2. Treatment with MFSCE at all concentrations (1, 2.5, and 5 µg/mL) significantly inhibited APP protein expression, while MFSCE at a concentration of 5 µg/mL significantly reduced BACE and PS1 expression, compared to other concentrations. With respect to PS2, the MFSCE-treated group exhibited significant downregulation at all concentrations, and this inhibitory activity was the highest among all amyloidogenic proteins. Therefore, this result demonstrated that MFSCE has a protective effect against Aβ25–35-induced amyloidogenesis via the downregulation of APP, BACE, PS1, and PS2 protein expression in SH-SY5Y cells.

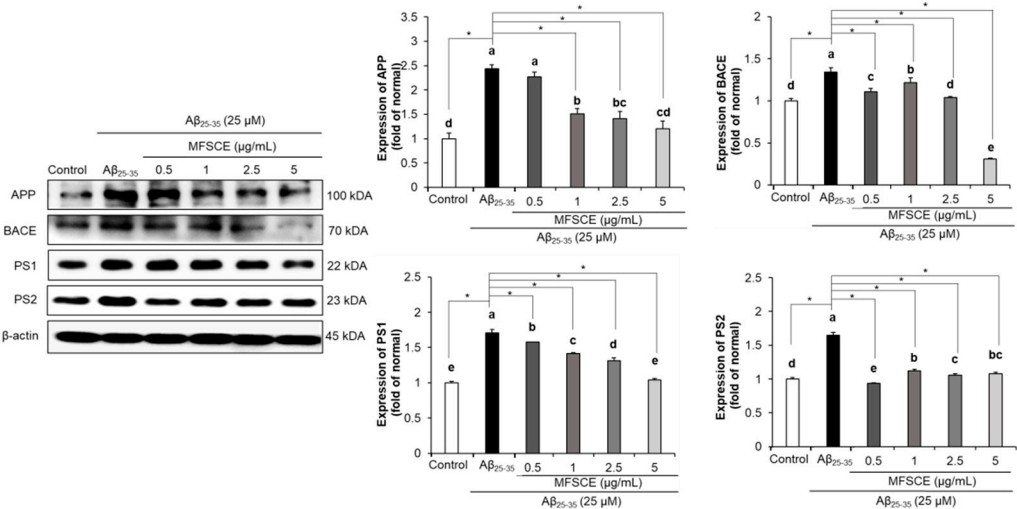

**Figure 6.** Effect of membrane-free stem cell extract (MFSCE) on expression of amyloidogenic pathway-related proteins in SH-SY5Y cells treated with Aβ25–35. β-actin was used as a loading control. Values are presented as mean ± SD (*n* = 6). The different letters (a–e) among groups represent significant differences determined by ANOVA and Duncan's multiple range test (*p* < 0.05). * *p* < 0.05 vs. Aβ25–35-treated cells.

## 4. Discussion

Stem cells have several favorable characteristics, such as ease of harvest for autologous transplantation, high proliferation rate, and ability to differentiate into multiple cell types including neural cells; therefore, they are used for treating several diseases [20,21]. Numerous bioactive molecules secreted by stem cells, such as cytokines, chemokines, and growth factors, regulate several physiological processes [22]. In addition, many studies have reported the neuroprotective effect of stem cells that exist in various organs and tissues, such as bone marrow, blood, and adipose tissue, owing to their antioxidant, anti-inflammatory, and anti-apoptotic properties [23–25]. Among various tissues, adipose tissue-derived stem cells caused the degradation of Aβ, thereby making them a useful agent for the treatment of AD [26]. However, stem cells do not survive for a long duration in the body and can cause potential side effects, including immune rejection, allergic reactions, and damage to healthy organs [22,27]. In particular, stem cell membrane contains human leukocyte antigens, major histocompatibility complex (MHC) I, and MHC II, which induce immune rejection during stem cell therapy due to their highly variable characteristics [28]. Therefore, to overcome the limitations of stem cells, MFSCE was prepared by removing the membranes of adipose tissue-derived stem cells. The purpose of this study was to evaluate the neuroprotective effects of MFSCE on neurotoxicity in SH-SY5Y cells-treated with Aβ25–35.

$A\beta_{25–35}$ is a divided fragment obtained from $A\beta_{1–40}$, a highly toxic fragment that causes neurotoxicity in AD [29]. The SH-SY5Y treated with $A\beta_{25–35}$ cell model is widely used to evaluate the neuroprotective ability of various materials, including stem cells [30,31]. LDH is an intracellular enzyme, and the release of LDH is an indicator of cell death in SH-SY5Y neuronal cells treated with $A\beta_{25–35}$ [32]. $A\beta_{25–35}$ concentration to induce neurotoxicity in SH-SY5Y cells has been carried out by previous studies. The study was conducted from 5 to 50 μM of $A\beta_{25–35}$. The treatment of 25 μM $A\beta_{25–35}$ significantly decreased cell viability by about 60%, which was the proper cell viability to the evaluate protective effect of samples from neurotoxicity [33–35]. In this study, treatment with $A\beta_{25–35}$ inhibited cell viability and increased LDH activity, compared with the control cells, indicating $A\beta_{25–35}$-induced neurotoxicity. However, treatment with MFSCE attenuated neurotoxicity by increasing cell viability and inhibiting LDH-release activity. A previous study reported that adipose tissue-derived stem cells exhibit a neuroprotective effect by restoring the viability of oxidative stress-induced SH-SY5Y cells [36]. In addition, conditioned medium of adipose tissue-derived stem cells inhibited LDH-release activity in cortical neuronal cells treated with glutamate, which is suggestive of a neuroprotective effect [37]. Our results also indicated that the neuroprotective effect of MFSCE on neurotoxicity led to an increase in cell viability and inhibition of LDH activity in neuronal cells.

Oxidative stress is caused by an imbalance between the ROS level and the antioxidant system [38]. $A\beta_{25–35}$ causes oxidative stress by increasing the ROS level in the brain, resulting in lipid peroxidation, neuroinflammation, and cognitive impairment [38–40]. In addition, both the accumulation of Aβ plaques and over-production of ROS were observed in the brains of patients with AD [38,39]. Therefore, many studies have been conducted to reduce oxidative stress, which will facilitate AD treatment by decreasing ROS production [40]. Our findings indicated that MFSCE significantly reversed the over-production of ROS in neuronal cells treated with $A\beta_{25–35}$. A previous study also reported that adipose tissue-derived stem cells ameliorated oxidative stress by inhibiting the ROS level and enhancing ROS scavenging ability to repair oxidized molecules in the hippocampus of APP/PS1 transgenic AD mice [41]. Our results also supported the fact that MFSCE derived from adipose tissue attenuated oxidative stress by decreasing the aROS level in neuronal cells induced by $A\beta_{25–35}$.

Neurotoxicity induced by $A\beta_{25–35}$ accumulation activates the release of inflammatory mediators [6]. Both iNOS and COX-2 activate nuclear factor-κB (NF-κB) signaling, which is a key regulator of inflammatory reactions in AD [42]. When this pathway is activated, inflammatory mediators stimulate the release of inflammatory cytokines, including interleukin (IL)-6, tumor necrosis factor-α (TNF-α), and IL-1β [6,42]. To investigate the molecular mechanisms underlying the effect of MFSCE on neuroinflammation, we measured the protein levels of inflammatory mediators, including iNOS and COX-2. Treatment with MFSCE dose-dependently downregulated iNOS and COX-2. Our previous study results indicated that MFSCE at doses of 1 and 2 μg/mL attenuated inflammatory reactions via the downregulation of iNOS and COX-2 protein expressions in LPS-induced RAW264.7 [14]. The conditioned medium of adipose tissue-derived stem cells inhibited inflammatory cytokines including IL-2, TNF-α, and IL-6 in inflammation-induced human astrocytes [43]. Based on these studies, we suggest that MFSCE exhibits the neuroprotective effect against neurotoxicity by downregulating inflammatory mediators in neuronal cells.

Neuronal apoptosis is regulated by various molecular mechanisms in the brains of patients with AD [44]. Bax and Bcl-2 are related to mitochondrial-mediated neuronal apoptosis [45]. When neurotoxicity is induced by excessive Aβ accumulation, the Bax level is increased, while the Bcl-2 level is decreased in patients with AD [46]. Therefore, Bax is considered a pro-apoptotic factor, while Bcl-2 is an anti-apoptotic factor. The MFSCE-treated SH-SY5Y cells showed a decreased ratio of Bax/Bcl-2 compared with that in $A\beta_{25–35}$-treated cells. High-concentration MFSCE (5 μg/mL)-treated cells markedly downregulated Bax expression, among other concentrations. In our previous study, treatment with MFSCE

(0.5, 1, and 2.5 μg/mL) decreased the ratio of Bax/Bcl-2 in oxidative stress-induced LLC-PK$_1$ cells [15]. In addition, adipose tissue-derived stem cells decreased the ratio of Bax/Bcl-2 in hepatic cytotoxicity-induced liver tissue [47]. Similarly, our results also demonstrated that MFSCE decreased the ratio of Bax/Bcl-2 in neuronal cells induced by Aβ$_{25-35}$. We suggest that MFSCE exhibits the neuroprotective effect by attenuating neuronal apoptosis.

During the development of AD, Aβ is generated via the activation of the amyloidogenic pathway [3,48]. In the first step of Aβ production, APP is cleaved to the β-secretase C-terminal fragment (β-CTF) by the β-secretase enzyme, which has been identified as BACE [49]. Subsequently, β-CTF is further cleaved to Aβ by γ-secretase, which is composed of PS1 and PS2 [50]. In particular, these enzymes are the mediators of rate-limiting processing steps leading to the generation of Aβ from APP [48]. Therefore, many researchers have focused on the regulation of the amyloidogenic pathway for the inhibition of Aβ generation as an effective strategy in the treatment of AD [48,51]. To elucidate the effect of MFSCE on the amyloidogenic pathway, we observed that MFSCE significantly downregulated levels of the amyloidogenic pathway-related proteins, such as APP, BACE, PS1, and PS2. Treatment with MFSCE at doses of 1, 2.5, and 5 μg/mL inhibited APP. MFSCE at a high concentration (5 μg/mL) effectively decreased BACE, compared with other concentrations. Moreover, cells treated with MFSCE dose-dependently downregulated PS1 expression. In addition, MFSCE at all concentrations (0.5–5 μg/mL) effectively downregulated PS2 among other amyloidogenic pathway-related proteins. Previous studies have reported the deposition of Aβ in stem cells. Adipose tissue-derived stem cells inhibited the accumulation of Aβ by stimulating Aβ-degrading enzyme, neprilysin, during Aβ metabolism [52]. In addition, human umbilical cord-derived stem cells decreased Aβ deposition and led to improved cognitive function in an AD mouse model [53]. Furthermore, we confirmed the neuroprotective effect of MFSCE derived from adipose tissue via the regulation of the amyloidogenic pathway in neuronal cells induced by Aβ$_{25-35}$.

This study investigated the effects of MFSCE against neurotoxicity in Aβ-induced SH-SY5Y cells, but it has several limitations, such as the comparison of neuroprotective effects between MFSCE and regular stem cells. Previous studies have demonstrated the neuroprotective effects of stem cells against Aβ-induced neurotoxicity in neuronal cells [10,11,54]. The treatment of stem cells increased cell viability and decreased Aβ levels, compared with Aβ-induced SH-SY5Y neuronal cells [11]. In addition, the treatment of stem cell attenuated the neuronal apoptosis by the regulation of the mitogen activated protein kinase signaling pathway in Aβ-induced hippocampal neuronal cells [10]. However, for the clinical application of disease therapy of MFSCE, further study on the comparison of the neuroprotective effects between MFSCE and stem cell has to be supported.

## 5. Conclusions

In the present study, MFSCE, a stem cell without a cell membrane, exerted a neuroprotective effect against neurotoxicity induced by Aβ$_{25-35}$. MFSCE attenuated oxidative stress by decreasing the ROS level. MFSCE attenuated neuroinflammation and neuronal apoptosis, and regulated amyloidogenic pathway-related proteins in SH-SY5Y cells treated with Aβ$_{25-35}$. Therefore, we propose MFSCE as a useful agent for AD.

**Author Contributions:** Conceptualization, E.J.C. and Y.S.K.; investigation, Q.Q.P.; writing—original draft preparation, H.S.P. and J.H.K.; writing—review and editing, J.H.K. and E.J.C.; supervision, E.J.C. All authors have read and agreed to the published version of the manuscript.

**Funding:** This research received no external funding.

**Institutional Review Board Statement:** Not applicable.

**Informed Consent Statement:** Not applicable.

**Data Availability Statement:** The data associated with this research are available and can be obtained by contacting the corresponding author.

**Conflicts of Interest:** The authors declare no conflict of interest.

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
