# Peer review of "Neuroprotective Effect of Membrane-Free Stem Cell Extract against Amyloid Beta 25–35-Induced Neurotoxicity in SH-SY5Y Cells"

_applsci, doi:10.3390/app11052219_

Round 1
Reviewer 1 Report
The manuscript by Park et al. regarding neuroprotective effects of MFSCE is an interesting in vitro work. The paper has experiments that are well though-out and simple, thus able to answer some posed questions. However, the work would benefit from addressing the following pints:
1) The manuscript needs to be English language copy edited. Many sentences are not written correctly making them hard to understand and some word choices are not the best.
2) If suggesting that MFSCE could be useful for treatment of AD, more information regarding this needs to be provided. This would include even some simple mechanistic studies, more functional assays, and simple in vivo experiments. Also, dosing, delivery methods, etc. should be at least discussed in addition to possible issues of MFSCE for therapeutic use.
3) How many cells are needed to produce the amounts of MFSCE used in the studies? Please include this information. How would this also relate to dosing in animals and potentially even in humans.
4) Could the effects of MFSCE be tested in another cell system?
5) What was the cell control for MFSCE? Would stem cells be even necessary? Could we use just fibroblast preparation or a preparation from another type of cell culture?
6) The statistical analysis of the data is a bit confusing. Why use letters to show statistical significance and using Duncan's multiple range? Why not just use multi-group ANOVA with significances showed in standard asterisks denoting the different significance levels? This is not well explained in the article.
Author Response
Thank you for the valuable comments on this paper. We considered the comments carefully and the manuscript has been revised according to the comments.
The manuscript by Park et al. regarding neuroprotective effects of MFSCE is an interesting in vitro work. The paper has experiments that are well though-out and simple, thus able to answer some posed questions. However, the work would benefit from addressing the following pints:
1) The manuscript needs to be English language copy edited. Many sentences are not written correctly making them hard to understand and some word choices are not the best.
; This paper has been revised and edited by a professional editor at editage. We attached certificate (No. PUSU_6209) of English editing.
2) If suggesting that MFSCE could be useful for treatment of AD, more information regarding this needs to be provided. This would include even some simple mechanistic studies, more functional assays, and simple in vivo experiments. Also, dosing, delivery methods, etc. should be at least discussed in addition to possible issues of MFSCE for therapeutic use.
; We investigated the cognitive improvement effects of MFSCE in the Alzheimer’s disease (AD) mouse model (data not shown, it will be published soon). In the behavior test, the intraperitoneal (i.p.) administration of MFSCE improved cognitive function, compared with AD mouse model (data not shown). In addition, administration of MFSCE attenuated oxidative stress-, inflammation-, and apoptosis-related markers in the brain tissue. Therefore, we suggest that MFSCE could be a useful therapy for treatment of AD. Furthermore, MFSCE has been applied to hair loss in patients with androgenetic alopecia twice every day for 16 weeks (Lee et al., 2019). But, further clinical study on cognitive improvement effects of MFSCE is necessary for treatment of AD.
[References]
Tak, Y.J.; Lee, S.Y.; Cho, A.R.; Kim, Y.S. A randomized, double-blind, vehicle-controlled clinical study of hair regeneration using adipose-derived stem cell constituent extract in androgenetic alopecia. Stem Cells Transl. Med. 2020, 1-11.
3) How many cells are needed to produce the amounts of MFSCE used in the studies? Please include this information. How would this also relate to dosing in animals and potentially even in humans.
; The MFSCE was produced by T-stem’s patent technology (Korean patent application number 10-2014-0154754, 2014 Nov 07). In the preparation of MFSCE, the amounts of adipose tissue stem cells are commercially confidential, thereby this explanation has limitations. In our study, we investigated the protective effects of MFSCE at concentration of 0.5 – 5 μg/mL against neurotoxicity in Aβ25-35-induced SH-SY5Y neuronal cells. When MFSCE at various concentrations (0.5 – 20 μg/mL) was treated to SH-SY5Y cells, MFSCE at concentrations up to 5 μg/mL did not exhibit significant cytotoxicity. Therefore, in this study, MFSCE at 0.5 – 5 μg/mL has been used for investigation of neuroprotective effects without cytotoxicity. In addition, we further investigated the cognitive improvement effects of MFSCE under in vivo Alzheimer’s disease (AD) mouse model. The administration of MFSCE at dose of 100 mg/kg/day attenuated cognitive impairment such as novel object recognition, spatial memory ability, and learning and memory ability in Aβ25-35-induced AD mouse model (data not shown). Based on the in vivo study and human equivalent dose by Km factor (Nair and Jacob, 2016), we suggest that the concentration of 8.1 mg/kg/day of MFSCE can be applied for humans. In addition, to clarify the therapeutic agents of MFSCE, cognitive improvement effects of MFSCE in the human have to be studied.
[References]
Nair, A.B.; Jacob, S. A simple practice guide for dose conversion between animals and human. J. Basic Clin. Pharm. 2016, 7, 27.
4) Could the effects of MFSCE be tested in another cell system?
; We previously investigated that the effects of MFSCE in the various cellular system. The MFSCE inhibited inflammation in the rat primary chondrocytes and Raw 264.7 macrophage cells (Lee et al., 2019; Saralamma et al., 2019). In addition, treatment of MFSCE exhibited protective effect from oxidative stress by regulation of inflammation and apoptosis in the hydrogen peroxide-induced human periodontal ligament fibroblasts and 3-morpholinosydnonimine-treated LLC-PK1 renal proximal tubule cells (He et al., 2019; Kim et al., 2019). Furthermore, MFSCE promotes glucose uptake by regulation of insulin signaling and AMPK pathway in the 3T3-L1 cells (Kim et al., 2019).
[References]
Lee, H.J.; Lee, S.M.; Moon, Y.G.; Jung, Y.S.; Lee, J.H.; Saralamma, V.G.S.; Kim, Y.S.; Pak, J.E.; Lee, H.J.; Kim, G.S.; Heo, J.D. Membrane-free stem cell components inhibit interleukin-1α-stimulated inflammation and cartilage degradation in vitro and in vivo: a rat model of osteoarthritis. Int. J. Mol. Sci. 2019, 20, 4869.
Saralamma, V.V.G.; Vetrivel, P.; Kim, S.M.; Ha, S.E.; Lee, H.J.; Lee, S.J.; Kim, Y.S.; Pak, J.E.; Lee, H.J.; Heo, J.D.; Kim, G.S. Proteome profiling of membrane-free stem cell components by nano-LS/MS analysis and its anti-inflammatory activity. Evid. Based Complement Alternat. Med. 2019, 2019, 4683272, doi: 10.1155/2019/4683272.
He, M.T.; Kim, J.H.; Kim, Y.S.; Park, H.S.; Cho, E.J. Protective effects of membrane-free stem cell extract from H2O2-induced inflammation responses in human periodontal ligament fibroblasts. Journal of the Korea Academia-Industrial cooperation Society, 2019, 20, 95-103.
Kim, M.J.; Kim, J.H.; Park, H.S.; Kim, Y.S.; Cho, E.J. Protective effects of membrane-free stem cell extract against oxidative stress in LLC-PK1 cells. Journal of the Korea Academia-Industrial cooperation Society, 2019, 20, 303-312.
Kim, J.H.; Kim, M.J.; Park, H.S.; Kim, Y.S.; Cho, E.J. Membrane free stem cell extract from adipose tissue enhances glucose uptake in 3T3-L1 cells. J. Korean Med. Obes. Res. 2019, 19, 89-96.
5) What was the cell control for MFSCE? Would stem cells be even necessary? Could we use just fibroblast preparation or a preparation from another type of cell culture?
; The MFSCE was produced using T-stem’s patent technology (Korean patent application number. 10-2014-0154754, 2014 Nov 07). The human adipose tissue was provided by a healthy female with 2-degree obesity. The adipose tissue was obtained after the blood tests (hepatitis B virus, hepatitis C virus, human T lymphocytic virus, human immunodeficiency virus, parvovirus B19, cytomegalo-virus, Epstein–Barr virus, and Treponema Pallidum) and doctor’s diagnosis, which proven to be appropriate. The donor gave written informed consent, and the Regional Committee on Biomedical Research Ethics approved the clinical protocol. The human adipose tissues were separated and purified, and extracted cells were cultured with a serum free cell culture medium in 5% CO2 incubator at 37°C. When the cells were confluenced, cells were subcultured. To removal of cell membrane, cells were collected and ultrasonificated. And then, the debris was eliminated from the membranes by centrifugation at 800 – 1500 g, following successive filtration. MFSCE, the final product, is proven to be a nontoxic substance through nine safety tests at the Good Laboratory Practice (GLP) accreditation authority.
6) The statistical analysis of the data is a bit confusing. Why use letters to show statistical significance and using Duncan's multiple range? Why not just use multi-group ANOVA with significances showed in standard asterisks denoting the different significance levels? This is not well explained in the article.
; Duncan’s multiple test is widely used in comparison various groups. The different letters indicated significant differences among groups, thereby same letters are statistically not significant. In addition, based on statistical analysis using Duncan’s multiple test, we additionally added the asterisks denoting the different significance levels.

Reviewer 2 Report
The focus of the manuscript by Park et al. entitled “Neuroprotective Effects of Membrane-Free Stem Cell Extract 2 against Amyloid Beta25-35-Induced Neurotoxicity in SH-SY5Y 3 Cells” is on the search for efficient and safe neuroprotectant against amyloid-Induced toxicity. The authors employed their broad experience with membrane-free stem cell extracts (MFSCE) and tested it using in vitro cellular system. Although the scope of the presented study is rather modest (only one cell line tested and a few basic molecular mechanisms studied), some interesting and promising results make the whole story worth publishing. However, some issues listed below should be considered by the authors:
- Although the “Introduction” provides the reader with appropriate background of the study, it will do even better with more up to date literature references (e.g. it seems obvious that a lot happened in the field since publishing [9])
- How the Aβ25-35 solution was obtained (source of the peptide) and handled? Such peptides are usually prone to aggregation – did the authors monitor its behavior in solution?
- Some further methodological details should be included: donors of human adipose tissue; source of ECL solution; number of repetitions used to calculate statistics; were the protein signal intensities in Western Blots normalized to actin signal (this may be critical in the context of less evident differences in protein levels shown e.g. in Fig. 5)?
- The authors should briefly discuss the reasons behind adjusting concentration of Aβ25-35 to 25uM
- The sentence in l. 345 (“ Several…”) is not fully clear – the authors could try to rephrase it.
- Abbreviations should be defined by their first usage (e.g. LDH)
- The whole text should be carefully cross-checked for typing errors (e.g. l. 43, 80 etc.) and English grammar by the authors and a native speaker.
Author Response
The focus of the manuscript by Park et al. entitled “Neuroprotective Effects of Membrane-Free Stem Cell Extract 2 against Amyloid Beta25-35-Induced Neurotoxicity in SH-SY5Y 3 Cells” is on the search for efficient and safe neuroprotectant against amyloid-Induced toxicity. The authors employed their broad experience with membrane-free stem cell extracts (MFSCE) and tested it using in vitro cellular system. Although the scope of the presented study is rather modest (only one cell line tested and a few basic molecular mechanisms studied), some interesting and promising results make the whole story worth publishing. However, some issues listed below should be considered by the authors:
Thank you for the valuable comments on this paper. We considered the comments carefully and the manuscript has been revised according to the comments.
- Although the “Introduction” provides the reader with appropriate background of the study, it will do even better with more up to date literature references (e.g. it seems obvious that a lot happened in the field since publishing [9])
; We revised the literature references (ref 2 and 9) in introduction section.
- How the Aβ25-35 solution was obtained (source of the peptide) and handled? Such peptides are usually prone to aggregation – did the authors monitor its behavior in solution?
; The Aβ peptide is easily aggregated and it formed Aβ plaque in the brain (DeTure and Dickson, 2019). In particular, Aβ plaque is observed in brain of AD patients, therefore it is widely known as a hallmarker of AD (Takahashi et al., 2017). Therefore, to induce neurotoxicity of Aβ plaque, Aβ was dissolved in saline solution and it was incubated at 37°C for 72 h before each experiments (Mairuae et al., 2019). We explained the method of the preparation of Aβ25-35 solution in materials and methods section (Page 2).
[Materials and Methods]
Aβ25-35 was purchased from Sigma-Aldrich (St. Louis, MO, USA) and it was incubated at 37°C for 72 h in saline solution. After aggregation of Aβ25-35, we used it in experiments.
[References]
Takahashi, R. H.; Nagao, T.; Gouras, G. K. Plaque formation and the intraneuronal accumulation of β‐amyloid in Alzheimer's disease. Pathology international, 2017, 67, 185-193.
DeTure, M.A.; Dickson, D.W. The neuropathological diagnosis of Alzheimer's disease. Mol Neurodegener. 2019, 14, 32.
Mairuae, N.; Connor, J.R.; Buranrat, B.; Lee, S.Y. Oroxylum indicum (L.) extract protects human neuroblastoma SH‑SY5Y cells against β‑amyloid‑induced cell injury. Mol. Med. Rep. 2019, 20, 1933-1942.
- Some further methodological details should be included: donors of human adipose tissue; source of ECL solution; number of repetitions used to calculate statistics; were the protein signal intensities in Western Blots normalized to actin signal (this may be critical in the context of less evident differences in protein levels shown e.g. in Fig. 5)?
; We revised it in materials and methods section ( Page 2-4).
[Materials and Methods]
The donors of human adipose tissue is healthy female in her twenties with 2-degree obesity. The adipose tissue was obtained after the blood tests, which proven to be appropriate.
Enhanced chemiluminescence (ECL) substrate solution was purchased from Bio-Rad Laboratories (Clarity Western ECL Substrate kit, Bio-Rad Laboratories, Inc Hercules, CA, USA).
Each experiment was carried out in triplicate (n = 3).
The protein level was expressed as a ratio of the band intensity divided by that of the housekeeping protein, β-actin.
- The authors should briefly discuss the reasons behind adjusting concentration of Aβ25-35 to 25uM
; We discussed it in discussion section (Page 8).
[Discussion]
The Aβ25-35 concentration to induce neurotoxicity in SH-SY5Y cells was determined by previous studies. The study was carried out from 5 uM to 50 uM of Aβ25-35. The treatment of 25 μM Aβ25-35 significantly decreased cell viability about 60%, that was proper cell viability to evaluate protective effect of samples from neurotoxicity (Jia et al., 2020; Xu et al., 2019; Wang et al., 2009).
[Reference]
Jia, G.; Yang, H.; Diao, Z.; Liu, Y.; Sun C. Neural stem cell conditioned medium alleviates Aβ25-35 damage to SH-SY5Y cells through the PCMT1/MST1 pathway. Eur. J. Histochem. 2020, 64, 3135.
Xu, H.N.; Li, L.X.; Wang, Y.X.; Wang, H.G.; An, D.; Heng, B.; Liu, Y.Q. Genistein inhibits Aβ25-35-induced SH-SY5Y cell damage by modulating the expression of apoptosis-related proteins and Ca2+ influx through ionotropic glutamate receptors. Phytother. Res. 2019, 33, 431-441.
Wang H, Xu Y, Yan J, Zhao X, Sun X, Zhang Y, Guo J, Zhu C. Acteoside protects human neuroblastoma SH-SY5Y cells against beta-amyloid-induced cell injury. Brain Res. 2009, 1283, 139-147.
- The sentence in l. 345 (“ Several…”) is not fully clear – the authors could try to rephrase it.
; We revised this sentence (Page 10).
- Abbreviations should be defined by their first usage (e.g. LDH)
; We defined full name of LDH at first usage (Page 2).
- The whole text should be carefully cross-checked for typing errors (e.g. l. 43, 80 etc.) and English grammar by the authors and a native speaker.
; This paper has been revised and edited by a professional editor at editage. We attached certificate (No. PUSU_6209) of English editing.

Reviewer 3 Report
The authors demonstrate the protective properties of the pre-treatment of SH-SY5Y neurons with membrane-free stem cell extract (MFSCE) in an in vitro model of Alzheimer’s disease. In this work the authors showed that the pre-treatment with MFSCE decreases the production of pro-inflammatory reactive oxygen species and increases cell viability through the inhibition of cell death. Also, the authors present data that suggest that the pre-incubation of Aβ25-35-treated SH-SY5Y neurons with MFSCE may decrease the formation of amyloid plaques in vitro. The manuscript presents many grammatical errors that decrease readability and must be improved. Also, the statistic approach used is not very clear. The discussion could be more extended as it is mostly a compilation of the data described in the manuscript with few integration.
The theme of the manuscript is interesting and new, opening possibilities for the study of new therapeutic approaches for neurodegenerative disorders.
Broad comments:
The entire manuscript requires extensive editing of English language.
The identification of the groups is not clear. Please replace the labelling. Normal by control cells and control cells by Aβ25-35 treated cells.
Indicate the number of independent experiments performed in the figure legend.
To use a one-way ANOVA, was the normal distribution of the data confirmed? Please clarify the statistic notation in the figure legends, so it is clear what columns present statistically significant differences or perform other more appropriate post test like Sidak's multiple comparisons test or Bonferroni multiple comparisons test. If not justify the use of Duncan’s multiple range test.
Provide the molecular weight of the bands observed in the western blot images.
Minor comments:
If the cell treatment was the same in all experiments add a cell treatment section to the methods and remove the details related to cell treatment from the other experimental detailing paragraphs.
Indicate antibody dilutions used in the methods section.
Define LDH.
Author Response
Thank you for the valuable comments on this paper. We considered the comments carefully and the manuscript has been revised according to the comments.
The authors demonstrate the protective properties of the pre-treatment of SH-SY5Y neurons with membrane-free stem cell extract (MFSCE) in an in vitro model of Alzheimer’s disease. In this work the authors showed that the pre-treatment with MFSCE decreases the production of pro-inflammatory reactive oxygen species and increases cell viability through the inhibition of cell death. Also, the authors present data that suggest that the pre-incubation of Aβ25-35-treated SH-SY5Y neurons with MFSCE may decrease the formation of amyloid plaques in vitro. The manuscript presents many grammatical errors that decrease readability and must be improved. Also, the statistic approach used is not very clear. The discussion could be more extended as it is mostly a compilation of the data described in the manuscript with few integration.
The theme of the manuscript is interesting and new, opening possibilities for the study of new therapeutic approaches for neurodegenerative disorders.
Broad comments:
The entire manuscript requires extensive editing of English language.
; This paper has been revised and edited by a professional editor at editage. We attached certificate (No. PUSU_6209) of English editing.
The identification of the groups is not clear. Please replace the labelling. Normal by control cells and control cells by Aβ25-35 treated cells.
; We replaced the label in the whole manuscript.
Indicate the number of independent experiments performed in the figure legend.
; We added the number of independent experiments in the Figure legend.
To use a one-way ANOVA, was the normal distribution of the data confirmed? Please clarify the statistic notation in the figure legends, so it is clear what columns present statistically significant differences or perform other more appropriate post test like Sidak's multiple comparisons test or Bonferroni multiple comparisons test. If not justify the use of Duncan’s multiple range test.
; According to the reviewer’s comment, statistical analysis was rechecked and the manuscript was revised. We revised it in materials and methods section (Page 4) and Figure legends.
[Materials and Methods]
The normal distribution of data was investigated using the Shapiro-Wilk test. When the group means of the data showed normal distribution, the results were analyzed by one-way analysis of variance (ANOVA) followed by Duncan’s multiple range test. If the group means of the data was no normal distribution, the differences between the groups were analyzed using Mann-Whitney U-test.
Provide the molecular weight of the bands observed in the western blot images.
; We added the molecular weight (kDA) of proteins in Figure 4-6.
Minor comments:
If the cell treatment was the same in all experiments add a cell treatment section to the methods and remove the details related to cell treatment from the other experimental detailing paragraphs.
; The cell treatment was explained the materials and methods section and the same explanation of experimental method was removed (Page 3).
[Materials and Methods]
2.4. Cell treatment
The SH-SY5Y cells were seeded at a cell density of 2.5 × 105 cells/well in a 96 well microplate and incubated for 24 h. The MFSCE was added to the each well at different concentrations (0.5 – 5 μg/mL) and incubated for 4 h. And then, the cells were treated with 25 μM Aβ25-35, followed by incubation for 24 h.
Indicate antibody dilutions used in the methods section.
; We added the antibody dilutions in materials and methods section (Page 3).
[Materials and Methods]
Primary and secondary antibodies are as follows: β-actin (1:1000, Cell signaling Tech., Beverly, USA), cyclooxygenase (COX-2, 1:500, Calbiochem Co., San Diego, USA), inducible nitric oxide synthase (iNOS, 1:1000, Calbiobhem Co., San Diego, USA), APP (1:1000, Sigma Aldrich, St. Louis, MO, USA), β-site APP-cleaving enzymes (BACE, 1:1000, Cell signaling Tech., Beverly, USA), presenilin 1 (PS1, 1:1000, Cell signaling Tech., Beverly, USA), presenilin 2 (PS2, 1:1000, Cell signaling Tech., Beverly, USA), B-cell lymphoma 2 (Bcl-2, 1:1000, Abcam, Cambridge, MA, USA), Bcl-2 associated X (Bax, 1:1000, Cell signaling Tech., Beverly, USA), anti-rabbit IgG, HRP-linked antibody (1:500, Cell signaling Tech., Beverly, USA).
Define LDH.
; We defined LDH ( Page 3; Page 8).

Reviewer 4 Report
In the present study Park et al propose MFSCE, stem cell without cell membrane, asneuroprotective agents against neurotoxicity induced by Aβ25-35.
The paper is potentially interesting.
The main limitation is the novelty since MFSCE has been already proposed has neuroprotectant strategy.
I would suggest to make additional experiments, on the most relevant findings, to make a comparison between MSCE and regular stem cells. This would help the reader to understand the effective potential of using MSCE.
Author Response
In the present study Park et al propose MFSCE, stem cell without cell membrane, asneuroprotective agents against neurotoxicity induced by Aβ25-35. The paper is potentially interesting.
The main limitation is the novelty since MFSCE has been already proposed has neuroprotectant strategy. I would suggest to make additional experiments, on the most relevant findings, to make a comparison between MSCE and regular stem cells. This would help the reader to understand the effective potential of using MSCE.
; Thank you for the valuable comments on this paper. In the present study, we investigated the effects of MFSCE against neurotoxicity in Aβ-induced SH-SY5Y cells. Several previous studies reported the effects of stem cell against Aβ-induced neurotoxicity (Shin et al., 2014; Lee et al., 2010; Kappy et al., 2018). Co-culture with stem cells in Aβ-induced SH-SY5Y cells increased cell viability and decreased Aβ levels, compared with Aβ-induced cells (Shin et al., 2014). Stem cell attenuated the neuronal apoptosis by regulation of phosphorylation of ERK and CREB in Aβ-induced hippocampal neuronal cells (Lee et al., 2010). Furthermore, treatment of stem cells protected from cell death, increased cell viability, and decreased expression of inflammatory cytokine (Kappy et al., 2018). However, stem cell has several limitations for clinical approach. The antigen such as histocompat-ibility complex (MHC) I and MHC II on cell membrane has a high cellular immune rejection by the human immune system (He et al., 2014; Marks et al., 2017; Tullis et al., 2014). On the other hand, MFSCE is produced by removing cell membrane from stem cell, so immune rejection is rarely occurred. Therefore, we suggest that MFSCE is suitable for replacing stem cell in the treatment of AD. However, the clinical study on neuroprotective effects of MFSCE has to be supported.
[References]
Shin, J.Y.; Park, H.J.; Kim, H.N.; Oh, S.H.; Bae, J.S.; Ha, H.J.; Lee, P.H. Mesenchymal stem cells enhance autophagy and increase β-amyloid clearance in Alzheimer disease models. Autophagy, 2014, 10, 32–44.
Lee, J.K.; Jin, H.K.; Bae, J.S. Bone marrow-derived mesenchymal stem cells attenuate amyloid β-induced memory impairment and apoptosis by inhibiting neuronal cell death. Curr Alzheimer Res. 2010, 7, 540-548.
Kappy, N.S.; Chang, S.; Harris, W.M.; Plastini, M.; Ortiz, T.; Zhang, P.; Hazelton, J.P.; Carpenter, J.P.; Brown, S.A. Human adipose-derived stem cell treatment modulates cellular protection in both in vitro and in vivo traumatic brain injury models. J. Trauma Acute. Care Surg. 2018, 84, 745-751.
He, H.; Yiu, S.C. Stem cell-based therapy for treating limbal stem cells deficiency: A review of different strategies. Saudi J. Ophthalmol. 2014, 28, 188-194.
Marks, P.W.; Witten, C.M.; Califf, R.M. Clarifying stem-cell therapy’s benefits and risks. N. Engl. J. Med. 2017, 376, 1007-1009. doi: 10.1056/NEJMp1613723.
Tullis, G.E.; Spears, K.; Kirk, M.D. Immunological barriers to stem cell therapy in the central nervous system. Stem Cells Int. 2014, 2014, 507905.

Round 2
Reviewer 1 Report
Thank you addressing most of the points raised. The point regarding cell control (point 5) was not addressed in the manuscript nor in the letter. Please explain if other cells could be used or does it need to be stem cell preparation. Why or why not? How could this be tested?
Author Response
Thank you addressing most of the points raised. The point regarding cell control (point 5) was not addressed in the manuscript nor in the letter. Please explain if other cells could be used or does it need to be stem cell preparation. Why or why not? How could this be tested?
In this study, we evaluated neuroprotective effects of membrane free stem cell extract (MFSCE) from adipose tissue against neurotoxicity in Aβ-induced neuronal cells. The MFSCE used in this study was produced from adipose tissue-derived stem cells, and it can not be extracted from the other tissue or cells. Stem cells can obtain from various tissues including bone marrow, adipose tissue, muscle, dental pulp, and synovium (Fellows et al., 2016). Among various tissues-derived stem cells, adipose tissue-derived stem cells are easily obtained from adipose tissue at multiple sites, and it has multi-lineage differentiation ability (Oedayrajsingh-Varma et al., 2006; Woo et al., 2016). Therefore, adipose tissue-derived stem cells are widely used for clinical application (Mohamed-Ahmed et al., 2018; Pikuła et al., 2013). Further study has to be supported for the possibility of MFSCE from various tissues such as bone marrow, muscle, and the others.
[References]
Fellows, C.R.; Matta, C.; Zakany, R.; Khan, I.M.; Mobasheri, A. Adipose, bone marrow and synovial joint-derived mesenchymal stem cells for cartilage repair. Front. Genet. 2016, 7, 213.
Oedayrajsingh-Varma, M.J.; van Ham, S.M.; Knippenberg, M.; Helder, M.N.; Klein-Nulend, J.; Schouten, T.E.; Ritt, M.J.; van Milligen, F.J. Adipose tissue-derived mesenchymal stem cell yield and growth characteristics are affected by the tissue-harvesting procedure. Cytotherapy 2006, 8, 166–177.
Woo, D.H.; Hwang, H.S.; Shim, J.H. Comparison of adult stem cells derived from multiple stem cell niches. Biotechnol. Lett. 2016, 38, 751–759.
Mohamed-Ahmed, S.; Fristad, I.; Lie, S.A.; Suliman, S.; Mustafa, K.; Vindenes, H.; Idris, S.B. Adipose-derived and bone marrow mesenchymal stem cells: a donor-matched comparison. Stem Cell Res. Ther. 2018, 9, 168.
Pikuła, M.; Marek-Trzonkowska, N.; Wardowska, A.; Renkielska, A.; Trzonkowski, P. Adipose tissue-derived stem cells in clinical applications. Expert Opin. Biol. Ther. 2013, 13, 1357-1370.

Reviewer 4 Report
The authors did not address all previous criticisms. In particular, no experiments on the comparison between MSCE and regular stem cells, have been perfomed
Author Response
The authors did not address all previous criticisms. In particular, no experiments on the comparison between MSCE and regular stem cells, have been perfomed
; Thank you for the valuable comments on this paper. Although we did not carry out the comparison between MFSCE and regular stem cells on the protective activity from neurotoxicity, the previous studies demonstrated the neuroprotective effects of stem cells against Aβ-induced neurotoxicity in neuronal cells (Shin et al., 2014; Lee et al., 2010; Kappy et al., 2018). Treatment of stem cells increased cell viability and decreased Aβ levels, compared with Aβ-induced cells, in the SH-SY5Y neuronal cells treated with Aβ (Shin et al., 2014). In addition, treatment of stem cell attenuated the neuronal apoptosis by regulation of phosphorylation of ERK and CREB in Aβ-induced hippocampal neuronal cells (Lee et al., 2010). Furthermore, treatment of stem cells protected from cell death, increased cell viability, and decreased expression of inflammatory cytokine (Kappy et al., 2018). However, stem cell itself has several limitations for treatment of diseases because of cellular immune rejections by human immune system. On the other hand, MFSCE is produced by removing cell membrane from stem cell, therefore it can be easily applied for the clinical applications without immune rejections. In the present study, we investigated the protective effects of MFSCE on neurotoxicity based on the previous studies on the stem cells.
[References]
Shin, J.Y.; Park, H.J.; Kim, H.N.; Oh, S.H.; Bae, J.S.; Ha, H.J.; Lee, P.H. Mesenchymal stem cells enhance autophagy and increase β-amyloid clearance in Alzheimer disease models. Autophagy, 2014, 10, 32–44.
Lee, J.K.; Jin, H.K.; Bae, J.S. Bone marrow-derived mesenchymal stem cells attenuate amyloid β-induced memory impairment and apoptosis by inhibiting neuronal cell death. Curr Alzheimer Res. 2010, 7, 540-548.
Kappy, N.S.; Chang, S.; Harris, W.M.; Plastini, M.; Ortiz, T.; Zhang, P.; Hazelton, J.P.; Carpenter, J.P.; Brown, S.A. Human adipose-derived stem cell treatment modulates cellular protection in both in vitro and in vivo traumatic brain injury models. J. Trauma Acute. Care Surg. 2018, 84, 745-751.
